# Intercepting an avoided α-iminol rearrangement with a Petasis reaction for the synthesis of 2,3-diaryl substituted indoles

Hui-Min Zhu[1], Tong Lei[1], Zhi-Xin Liao[1], Jia-Chen Xiang ⓘ [1] ✉ & An-Xin Wu ⓘ [2] ✉

Rearrangement reactions are generally considered to be a rapid and synergistic intramolecular reconstructing process that is insensitive to intermolecular intruders. We report that α-iminol rearrangements could be strategically redirected by the interception of Petasis reactions, in the context of being avoided by strong electron-withdrawing groups on the migrative aryl units. 1,4-Aryl migration prevails over 1,2-aryl migration via forming a boron-ate complex. By leveraging this reactivity, we developed a regiospecific synthesis of unsymmetrically 2,3-diaryl substituted indoles from three readily available feedstocks: an amine, an arylglyoxal, and a boronic acid. While traditional Petasis reactions with similar three-component inputs are typically applied to build $C(sp^3)$-$C(sp^2)$ and $C(sp^3)$-$C(sp^3)$ bonds, the present transformation offers a special opportunity for constructing a $C(sp^2)$-$C(sp^2)$ linkage. Highly substituted indole motifs with structural diversity in the C2 position are easily accessed by this three-component reaction. A mechanism containing a copper-cobalt collaborative promotion process was suggested.

2,3-Disubstituted indoles are widely present in natural products and are displayed as privileged motifs in biologically active compounds[1–4]. As a special array, unsymmetrically 2,3-diaryl substituted indoles have received sustained attention due to their medicinal importance[5–7] and synthetic intractability[6–14]. As powerful named reactions, Fischer[6,15–17] and Cacchi[18–25] indole syntheses are the most versatile methods for constructing such skeletons, although their respective reactants need to be pre-synthesized. Pioneering work on transition-metal-catalyzed aniline-alkyne cyclization and their variants[26–39], led by Larock[26–28], Fagnou[29,30], and Glorius[31,32], respectively, provides efficient strategies for synthesizing 2,3-diaryl substituted indoles, despite their intrinsic difficulty in controlling the regioselectivity. In this respect, Yoshikai[40,41] and Wan[42] individually provided elegant options. Nevertheless, a regiospecific synthesis of unsymmetrical 2,3-diaryl-substituted indoles using noble-metal-free conditions and readily available starting materials remains valuable and desirable.

Recently, we reported an aerobic copper-catalyzed synthesis of 2,2-disubstituted indolin-3-ones using an amine, an arylglyoxal, and a nucleophile[43]. As a logical extension of this diverted Mannich reaction, we anticipated that using an aryl boronic acid, a masked aryl nucleophile, would

lead to the formation of 2,2-diaryl substituted indolin-3-ones based on a similar principle (Fig. 1a). However, beyond such routine extension, we predicted that it was possible to alter the domino sequence to afford a product with a completely distinct backbone utilizing identical substrates, therefore enriching the reaction diversity. Herein, we emphasize that an α-hydroxyiminium intermediate (e.g., **I-A** generated from the condensation of an amine and an arylglyoxal) is not only a precursor for α-iminol rearrangement[44–49] but also an ideal partner for the Petasis reaction[50–62]. Then a boron-ate complex **I-B** would intercept the original pathway and redirect the transformation into a 2,3-diaryl substituted indole (Fig. 1b). It is not surprising that transformations that switch between these two classical reactions have not been reported, since such a design faces two challenges: (A) Using an intermolecular reaction to intercept an intramolecular one is kinetically difficult. (B) Tactics for intervening in the 1,2-aryl migration which belongs to the α-iminol rearrangement are still unknown, as the method for avoiding such a rapid reconstructing process is underexplored[63–67]. Inspired by Cram's impactful study on phenonium ion theory in connection with Wagner-Meerwein rearrangements[68–71], we assessed that the *para* substitution on the migrative aryl units affects the

[1]School of Chemistry and Chemical Engineering, Southeast University, Nanjing, 211189, PR China. [2]State Key Laboratory of Green Pesticide, International Joint Research Center for Intelligent Biosensor Technology and Health, College of Chemistry, Central China Normal University, Wuhan, 430079, PR China.
✉e-mail: xiangjiachen@seu.edu.cn; chwuax@mail.ccnu.edu.cn

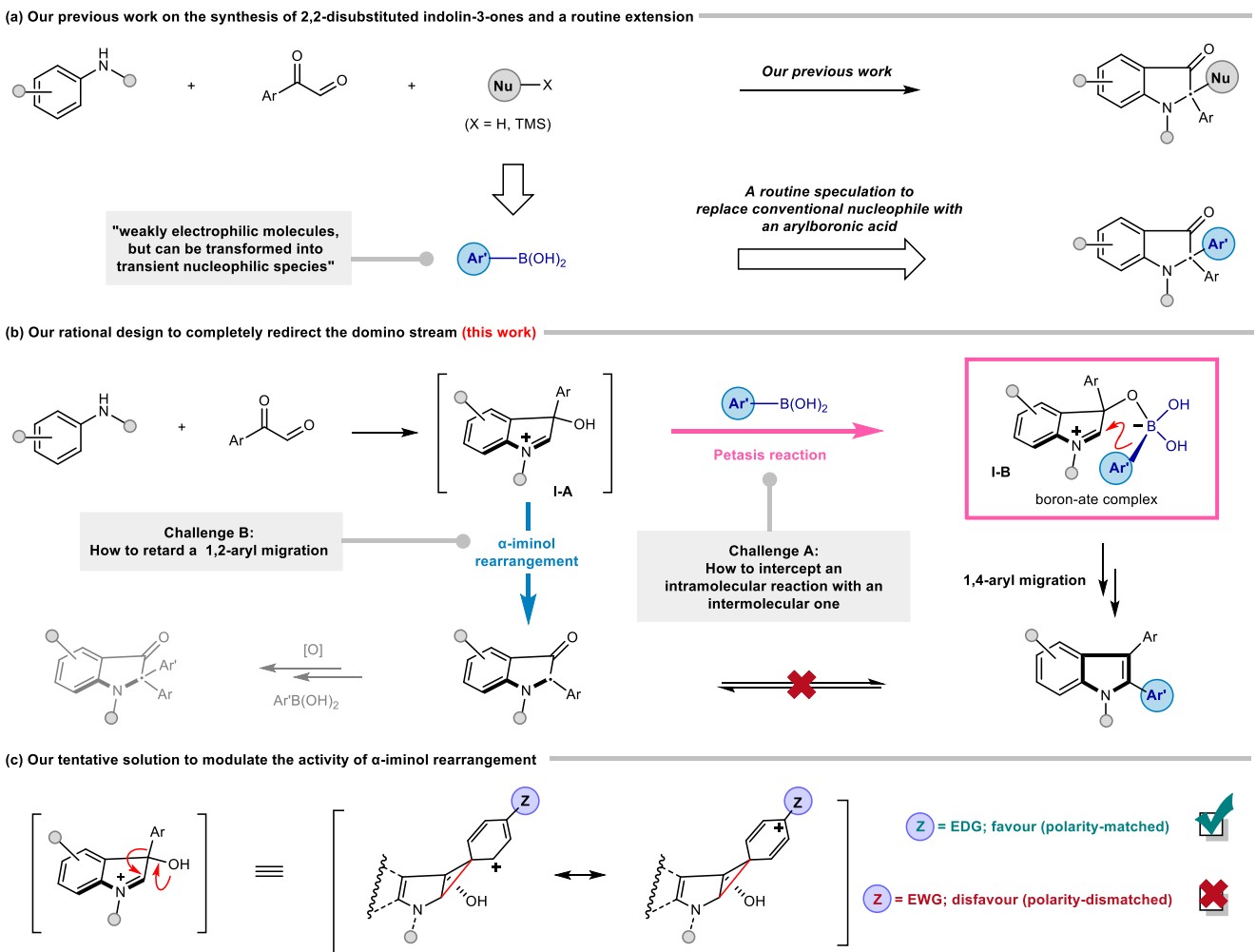

**Fig. 1 | A brief introduction of our previous work and the current working hypothesis. a** Our previous work on the synthesis of 2,2-disubstituted indolin-3-ones and a routine extension. **b** Our rational design to completely redirect the domino stream. **c** Our tentative solution to modulate the activity of α-iminol rearrangement.

migratory aptitude. In general, electron-donating groups stabilize the discrete carbon cation at its α- and conjugated γ-site, thus favoring such anionotropic rearrangement[72,73]. In sharp contrast, intermediate **I-A** bearing an aryl with an electron-withdrawing *para*-substitution would have a lower migrative reactivity but a longer lifetime due to its mismatched polarity, pending the collision of boric acid (Fig. 1c). Through 1,4-aryl migration, highly substituted indole motifs could be obtained after elimination. Moreover, Petasis reaction and its variants, using amines, aldehydes, and boric acids, are employed extensively for C(sp³)-C(sp²)[56–59] bond formation, as well as recently emerged C(sp³)-C(sp³)[60–62] bond connections. The application of this versatile named reaction to forge a C(sp²)-C(sp²) linkage remains elusive, thus further encouraging us to verify our hypothesis. Here, we report a three-component reaction of an arylamine, an arylglyoxal, and a boronic acid, achieving the selective synthesis of 2,3-diaryl substituted indoles and 2,2-diaryl substituted indolin-3-ones. Catalytic amount of copper trifluoroacetate hydrate and Co(salen) (II) are used as the reagents. More than 50 diverse examples are presented.

## Results

### Transformation developments

Through screening a series of variables and conditions (Please see Supplementary Table 1 on page S2 for details), we found that treating three commercially available materials: bis(4-methoxyphenyl)amine (**1a**), 4-nitrophenylglyoxal (**2a**) and phenylboronic acid (**3a**) with a catalytic manifold of copper trifluoroacetate hydrate (20 mol%) and Co(salen) (II) (20 mol%) in a solution of DCE (0.1 M) at 80 °C in the air atmosphere for

4 h led to the formation of the corresponding indole product (**4a**) in 56% yield (Table 1, entry 1). These optimum conditions are recognized as our standard conditions in the following sections. Notably, the replacement of **2a** with other counterparts bearing electron-withdrawing groups, such as 4-Cl, 4-CF₃, 4-CN, failed to afford the desired product **4**, but provided **4ac** type indolin-3-one product in varying rates of yield (entry 2). Those results indicated that only *para*-NO₂, which is the strongest electron-withdrawing group according to the Hammett constant[74], could effectively avoid α-iminol rearrangement to match intermolecular interception under our conditions. Replacement of copper trifluoroacetate hydrate with TFA also produced **4a**, despite reducing the yield (entry 4), suggesting that the copper salt acts as a Lewis acid. Using Cobaloxime(III) instead of Co(salen) (II) could still afford **4a** in a useful yield (entry 5). More controlled experiments demonstrated that this reaction is facilitated by copper and cobalt (entries 6-8). Although the role of cobalt is still unclear, we tentatively envisage that this cobalt-Schiff base might assist in the deprotonation process during the generation of the boron-ate complex through forming Co(salen)/O₂ adducts (Please see Supplementary Discussion on page S71 for details)[75–80]. We have also attempted to screen other additives besides cobalt, such as oxidants, bases, and additional Brønsted or Lewis acids, but have not been able to improve yields. Next, using a co-solvent of DCE and HFIP was unable to further improve the yield (entry 9), although it has been reported that HFIP can promote the Petasis process[81]. A relatively high temperature and an air atmosphere are beneficial (entries 10-11). Employing phenylboronic acid pinacol ester (PhBpin) or potassium phenyltrifluoroborate (PhBF₃K) instead of phenylboronic acid failed to produce **4a** (entry 12).

**Table 1 | Screening of reaction conditions.**[a]

| Entry | Variation from 'standard conditions' | Yield of 4 (%)[b] |
|---|---|---|
| 1 | none | 56 |
| 2 | Ar¹/Ar²/Ar³ instead of Ar⁴ | N.D |
| 3 | CuBr as a copper source | 17 |
| 4 | TFA instead of Cu(TFA)₂·xH₂O | 30 |
| 5 | Cobaloxime(III) instead of Co(salen) | 50 |
| 6 | without Co(salen) | 42 |
| 7 | without Cu(TFA)₂·xH₂O | 25 |
| 8 | without both Co(salen) and Cu(TFA)₂·xH₂O | 10 |
| 9 | DCE: HFIP = 5:1 as solvent | 31 |
| 10 | under 60 °C | 33 |
| 11 | under N₂ | 20 |
| 12 | PhBpin/PhBF₃K instead of 3a | N.D/N.D |

[a]Reaction conditions: **1a** (0.2 mmol), **2** (0.4 mmol, 2.0 equiv), **3a** (0.6 mmol, 3.0 equiv), Cu(TFA)₂·xH₂O (0.04 mmol, 0.2 equiv), Co(salen) (0.04 mmol, 0.2 equiv), DCE (2.0 mL, c = 0.1 M), 80 °C, 4 h, air atmosphere. [b]Isolated yield. N.D. Not detected.

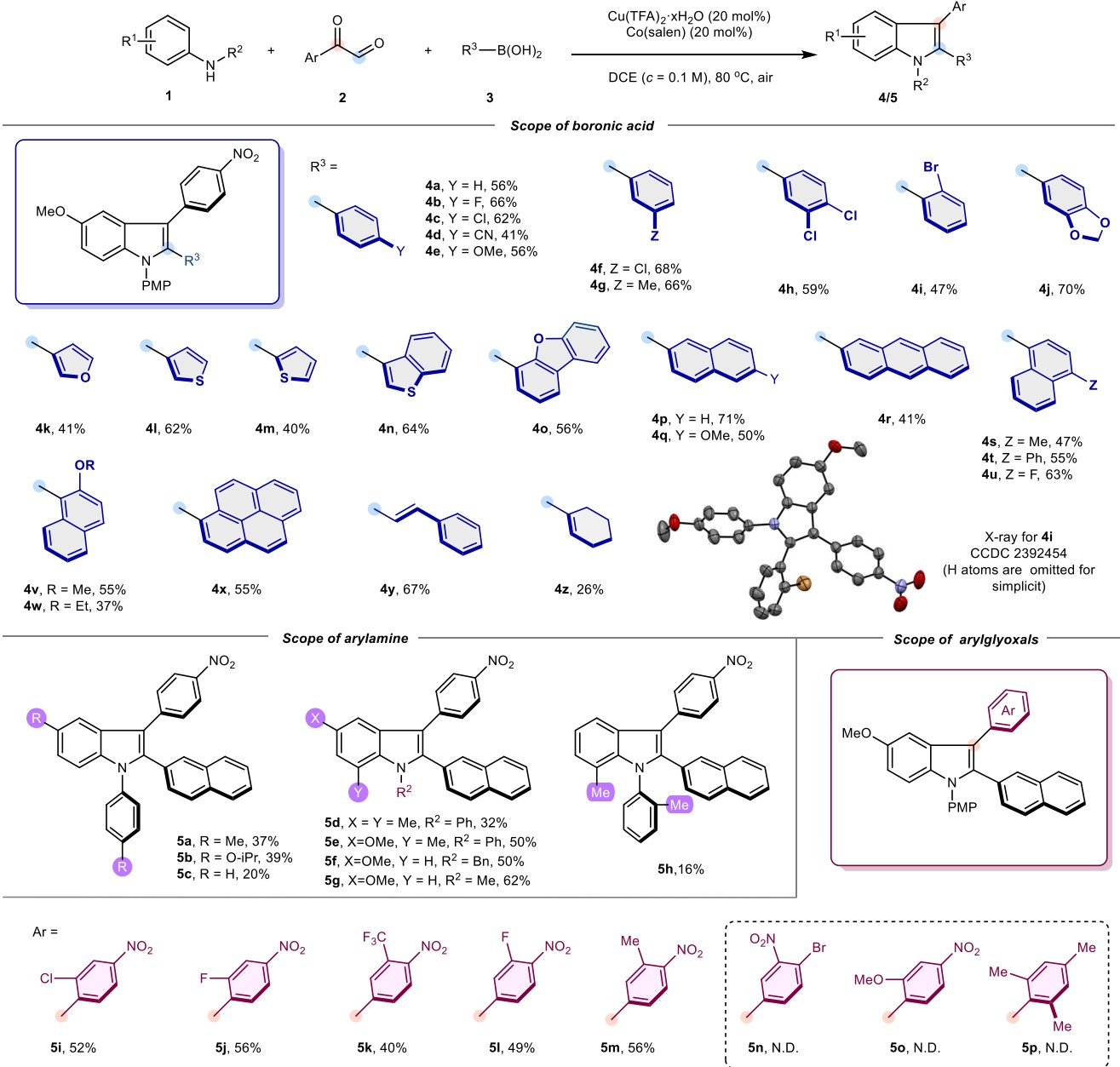

**Fig. 2 | Reaction scope for the synthesis of 2,3-diaryl substituted indoles.** Reaction conditions: **1** (0.2 mmol), **2** (0.4 mmol, 2.0 equiv), **3** (0.6 mmol, 3.0 equiv), Cu(TFA)$_2$·xH$_2$O (0.04 mmol, 0.2 equiv), Co(salen) (0.04 mmol, 0.2 equiv), DCE (2.0 mL, $c$ = 0.1 M), 80 °C, air atmosphere. Isolated yield.

These results suggest that the reaction process may not involve a metal insertion. It is worth mentioning that the regioisomer **4ab** and indolin-3-one byproduct **4ac** were not detected throughout the screening process. We also did not detect the classical linear Petasis product **4ad** during the optimization. This may be due to the lack of the adjacent -OH group in the **2a** substrate, compared to the conventional Petasis substrates (e.g., glyoxylic acid, glycolaldehyde, and salicylaldehyde)[56–62], therefore unable to activate the boric acid to trigger such a reaction.

## Synthesis of 2,3-diaryl substituted indoles by the reaction of amine, arylglyoxal, and a boronic acid

With the optimum conditions in hand, the scope of organoboronic acids was then examined (Fig. 2). Arylboronic acids with electron-donating substitutions (**4e, 4 g, 4j**), halogens (**4b, 4c, 4 f, 4i**) on either *ortho/para/meta* position, as well as di-halogens (**4 h**) performed good reactivity affording desired products in moderate to high yields. Electron-deficient arylboronic acids, such as 4-cyanophenyl, were also tolerated in the transformation, with

a relatively low yield. These reactive tendencies are consistent with previous reports of typic Petasis reactions concomitant with an aryl migration[82]. Furthermore, heteroaryl boronic acids bearing 3-furanyl, 2- or 3-thienyl rings were suitable substrates (**4k-4m**). Sterically hindered heterocycles with fused-ring systems, such as 3-benzothiophene, and 4-benzo[b,d]furan proceeded well in this reaction (**4n, 4o**). A range of boronic acids bearing polycyclic aromatic hydrocarbons, no matter in the form of C1- or C2-substituted, were well accommodated to afford indole motifs which are not easily accessed by other methods (**4p-4x**). To our delight, a high yield was observed when vinyl boronic acid was employed (**4 y**). Since the olefin motif can be easily reduced into a saturated one, this result further broadens the utility of the present indole synthesis. Finally, it was found that cyclohexene-1-boronic acid could also participate in the reaction, albeit with a low yield (**4z**). The substrate scope of arylamines as well as arylglyoxals appears to be more restricted compared to that of boronic acids. In general, at least one electron-donating group is required in aryl substituents of diarylamine for obtaining a better yield (**5a, 5b, 5 d, 5e**). When diphenylamine was used,

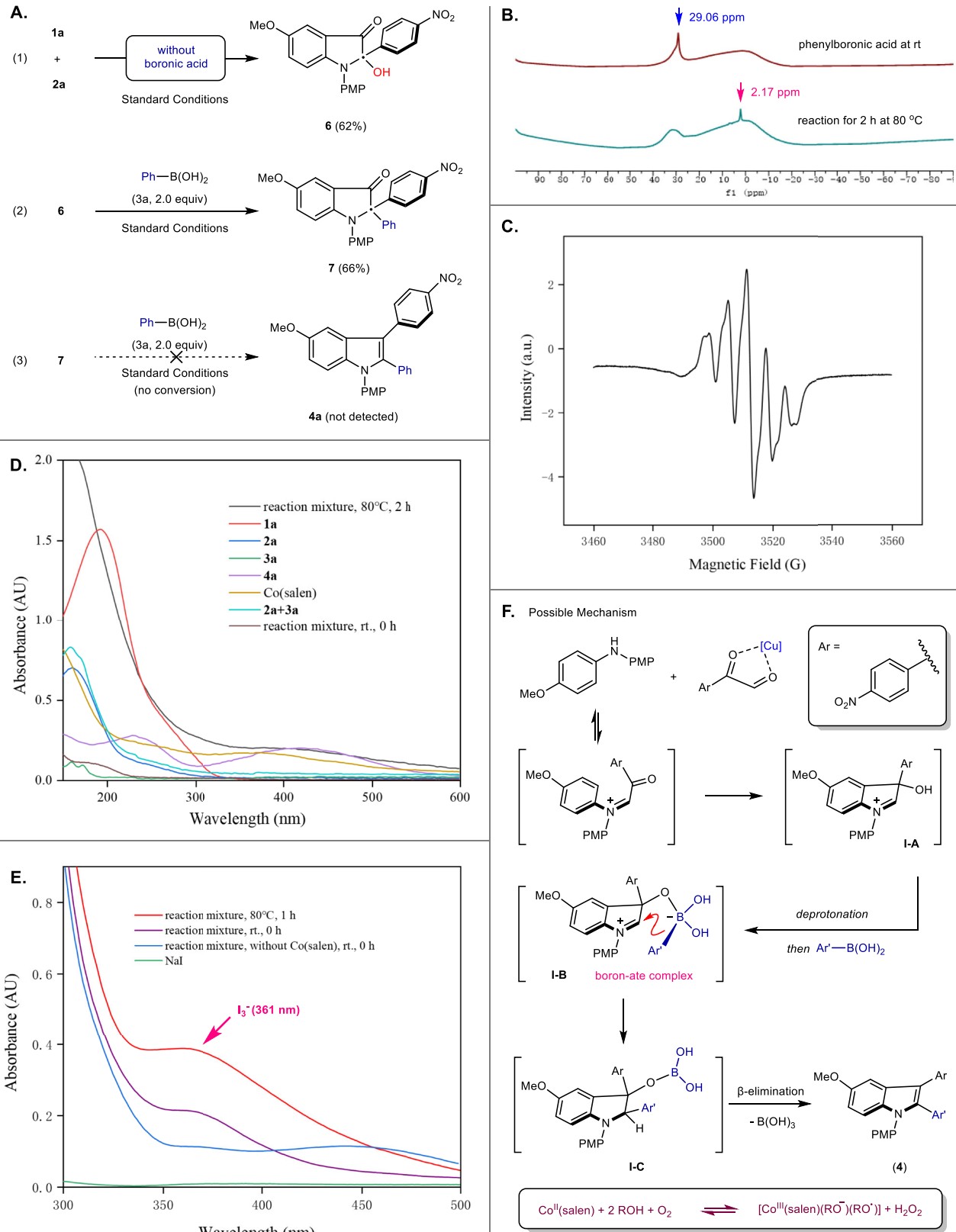

**Fig. 3 | Control experiments and mechanistic considerations. A** Control experiments. **B** [11]B NMR spectroscopy experiments. **C** EPR experiment. **D** UV-vis absorption spectra of different reaction components. **E** Measurement of the $H_2O_2$ in the reaction mixture. **F** Possible mechanism.

only 20% yield of the target product was obtained (**5c**). *N*-benzyl-4-methoxyaniline and 4-methoxy-*N*-methylaniline afforded the corresponding *N*-benzyl (**5 f**) and *N*-methyl (**5 g**) indole products, respectively. When di-*o*-tolylamine was submitted into the reaction, the corresponding product was

obtained in only 16% yield (**5 h**). Moreover, after extensive testing, we found that only arylglyoxal substrates with a nitro group attached in the *para* position of the phenyl ring could fulfill the activity requirement for this transformation. Bearing halogen (-Cl, -F) or electron-withdrawing group

(-CF$_3$) at *ortho/meta* positions of the *p*-nitrophenyl substituent can also realize the conversions smoothly (**5i-5l**). Interestingly, good reactivities were obtained when the methyl group was attached to *meta* positions of the *p*-nitrophenyl (**5 m**). In contrast, transformations were completely muted when 2-methoxy-4-nitrophenyl (**5o**) or even 3-nitro-4-bromophenyl glyoxal (**5n**) were employed. These results suggest that the electronic properties of the carbon linked to the indole C3 position are critical. If its electron density is enhanced (e.g., affected by an *ortho* methoxyl group), the α-iminol rearrangement reaction would not be avoided, thus unable to be intercepted by the Petasis reaction. Furthermore, 2,4,6-trimethyl phenyl glyoxal turned out to be an unsuitable substrate illustrating steric effects of arylglyoxal might not be crucial to avoid 1,2-aryl migration (**5p**). Finally, the structure of **4i** was determined by X-ray crystallographic analysis. Notably, the current method might have the synthetic potential to create indoles bearing a chiral C2- (**4i, 4 v, 4w**) or C3- (**5i, 5j**) aryl axis[25], if a suitable asymmetric catalytic condition is finely established.

## Control experiments and mechanistic considerations

Control experiments were then conducted to provide more insights into the reaction mechanism (Fig. 3A). Exposure of **1a** and **2a** under our standard conditions in the absence of phenylboronic acid (**3a**) afforded **6** in 62% yield (Fig. 3A-1). This reactivity is consistent with the experimental results reported in our previous work[43]. Next, treating **6** under the standard conditions in the presence of **3a** rendered **4ac**-type indolin-3-one **7** as a classic Petasis product[83] in 66% yield (Fig. 3A-2). Furthermore, stirring **7** with extra phenylboronic acids (**3a**) failed to produce **4a**, indicating neither **6** nor **7** was the intermediate of our model transformation (Fig. 3A-3)[84]. These results further support our original mechanistic speculations (Fig. 1b and c). [11]B NMR spectroscopy experiments were then carried out to further demonstrate the formation of boron-ate complex during the reaction. An obvious up-field shift (from 29.1 ppm to 2.2 ppm) indicated the tetracoordinated boronate species were likely to be generated as we expected (Fig. 3B)[85–87]. Furthermore, UV–vis absorption spectrometry of the reaction mixture indicated there was no definitive proof for the formation of the EDA complex between the boronate anion and *p*-nitrophenyl region during the

formation of the boron-ate complex (**I-B**) (Fig. 3D)[88]. These preliminary experimental results indicate that the successful occurrence of this domino process is due to the strong electron-withdrawing group slowing down the α-iminol rearrangement rather than forming a strong EDA complex that accelerates the Petasis process.

To get more insights into the mechanism, an EPR experiment was performed. An obvious signal was captured when 5,5-dimethyl-1-pyrroline N-oxide (DMPO) was employed as a radical scavenger suggests that our reaction may involve the formation of radicals (Fig. 3 C). Furthermore, determined spectroscopically by the iodimetry method, H$_2$O$_2$ should be produced by cobalt during the reaction (Fig. 3E) which suggests a two-electron, two-proton reduction of oxygen may be involve in the reaction. Furthermore, it is reported that Petasis reaction is more favorable when it undergoes a mechanism *via* a deprotonated ate complex[89]. Indeed, our density functional theory (DFT) calculations illustrated that the process of forming boron-ate complex from a deprotonated hydroxyl group is 36.9 kcal mol$^{-1}$ lower in energy than the process without deprotonation in this reaction (Please see Supplementary Fig. S7). According to these experimental results, a possible mechanism is suggested in Fig. 3F. The role of copper salt is to activate **1a** by coordination with the adjacent dicarbonyl group. Given that cobalt acts primarily as a reaction accelerator rather than an essential additive (Table 1, entry 6), we propose that the cobalt-oxygen synergy facilitates intermediate **I-A** deprotonation probably through in situ base generation via cobalt-mediated oxygen reduction[78–80]. We discuss these details more specifically in the Supplementary Discussion on page S71. At this stage, we are still unable to further explain or prove this mechanism, especially the role of cobalt. We hope to continue to explore the mechanism of action of this bimetallic reagent in our subsequent studies.

Based on our understanding of the reaction mechanism, we further adapted the transformation using arylglyoxal bearing non-strongly electron-withdrawing groups. 2,2-Diaryl substituted indolin-3-ones were successfully obtained under the identical reaction conditions otherwise in the absence of Co(salen) (Fig. 4). 2,2-Diaryl substituted indolin-3-ones are privileged motifs but not easy to access, especially for those with neither of the two aromatic rings bearing an electron-donating group[90–93].

**Fig. 4 | Synthesis of 2,2-diaryl substituted indolin-3-ones under modified conditions.** Reaction conditions: **1** (0.2 mmol), **2** (0.4 mmol, 2.0 equiv), **3** (0.6 mmol, 3.0 equiv), Cu(TFA)$_2$·xH$_2$O (0.04 mmol, 0.2 equiv), DCE (2.0 mL, *c* = 0.1 M), 80 °C, air atmosphere. Isolated yield. $^a$Reaction conditions for one-pot two stage protocol: **1** (0.2 mmol), **2** (0.4 mmol, 2.0 equiv), Cu(TFA)$_2$·xH$_2$O (0.04 mmol, 0.2 equiv), DCE (2.0 mL, *c* = 0.1 M), 80 °C, air atmosphere for 4 h, then **3** (0.6 mmol, 3.0 equiv) and additional Cu(TFA)$_2$·xH$_2$O (0.04 mmol, 0.2 equiv) were added, then react for 8 h at 80 °C, air atmosphere.

An elegant work was reported to synthesize this motif by Biju et al. just recently[94]. We are delighted to see that through slight adjustment of our additives, these valuable *N*-heterocyclic frameworks were obtained in a modular manner. It is noteworthy that 4-nitro-phenylglyoxals can be employed in the synthesis of the corresponding 2,2-diaryl substituted indolin-3-one (**8n**) through a one-pot, two-stage approach, whereby the arylglyoxals are allowed to react preferentially with the amine for 4 h, and then boric acid is added.

## Conclusions

In summary, we developed a regiospecific synthesis of unsymmetrically 2,3-diaryl substituted indoles by a multicomponent reaction of three easy-accessible linear substrates: an arylamine, an arylglyoxal, and a boronic acid. Facilitated by Cu(II), Co(II), and air, this transformation features operational simplicity and is suitable for a range of boronic acids bearing heteroaromatic rings, sterically hindered polyaromatic rings, as well as vinylic substitutions. A copper-cobalt additive manifold was developed to serve as an acid-base pair. This indole synthesis incorporates a Petasis process but with a rarely seen aryl-aryl bond formation, which provides an additional application scenario for this versatile named reaction. We also anticipate that our strategy using a robust intermolecular reaction to intercept an avoided intramolecular rearrangement would provide accessibility for other synthetically valuable targets, as well as expand the functionality of classical rearrangements from an updated perspective.

## Methods

In a 35 mL heavy-wall pressure tube were added 4,4′-dimethoxydiphenylamine **1a** (45.9 mg, 0.2 mmol), 4-nitrophenylglyoxal **2a** (71.7 mg, 0.4 mmol), boronic acid **3a** (73.2 mg, 0.6 mmol), Cu(TFA)$_2$·xH$_2$O (12.3 mg, 0.04 mmol), Co(salen) (13.1 mg, 0.04 mmol) and DCE (2.0 mL, *c* 0.1 M), and the resulting mixture was stirred at 80 °C (heating block) for 4 h until substrate conversion was almost complete by TLC analysis. After the reaction stopped and cooled at room temperature, the reaction mixture was quenched with saturated NaHCO$_3$ solution (50 mL) and NaCl solution (200 mL). The mixture was then extracted with EtOAc (150 mL × 2), and the organic layers were separated and merged. The mixture was dried with anhydrous Na$_2$SO$_4$ and concentrated under reduced pressure. The crude product was purified by column chromatography on silica gel (200–300 mesh, eluted with PE : DCM = 3:1) to afford the product **4a**.

## Data availability

The X-ray crystallographic coordinates for structures reported in this Article have been deposited at the Cambridge Crystallographic Data Centre (CCDC), under deposition numbers CCDC 2392454 (Supplementary Data 1) and 2411655 (Supplementary Data 2). These data can be obtained free of charge from The Cambridge Crystallographic Data Center via https://www.ccdc.cam.ac.uk/ structures/. All other data supporting the findings of the study, including experimental procedures and compound characterization, UV–vis absorption spectrometry, DFT calculations, EPR experiments are available within the paper and the Supplementary Information. Correspondence and requests for materials should be addressed to J.C.X, all data are available from the corresponding author upon request.

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

## Acknowledgements
This work was supported by the National Natural Science Foundation of China (Grant No. 22301035, 22171098, 32370063). This work was also supported by Jiangsu Province Youth Science and Technology Talent Support Project (JSTJ-2024-454), Southeast University's "Zhi-Shan Young Scholars of Excellence" Support Program (2242025RCB0021), Chengdu Guibao Science & Technology Co., Ltd and the 111 Project B17019. We are grateful to Prof. Pan Xu from Southeast University for valuable discussions.

## Author contributions
J.-C.X., W.-A.X. and Z.-X.L. conceived and designed the experiments. J.-C. X.H.-M.Z. and T.L. carried out the experiments. All the authors interpreted the results and co-wrote the manuscript.

## Competing interests
The authors declare no competing interests.
