## [Transparent Peer Review file · Communications Chemistry]

Intercepting an avoided α -iminol rearrangement with a Petasis reaction for the synthesis of 2,3-diaryl substituted indoles

Corresponding Author: Professor An-Xin Wu

Version 0:

Reviewer comments:

Reviewer #1

(Remarks to the Author)

This manuscript by Wu and co-workers describes an interesting synthesis of 2,3-diaryl indoles using an aniline, a glyoxal, and a boronic acid. The authors have previously reported the synthesis of indolinones in a similar manner by condensation of anilines with glyoxals, followed by a typical 1,2-aryl shift to form the final carbonyl. Here, it is found that when the aryl group is very electron-deficient (p-nitro), that aryl shift is very slow, presumably due to disfavored phenonium formation, and the iminol intermediate is therefore long-lived enough to undergo Petasis-type addition to form the indole products that are the subject of the study. Some mechanistic discussion is provided, although the insight is somewhat limited by the complexity of the system, involving two first-row metal catalysts, and though the scope of the reaction is obviously limited, the authors do explore what happens with less electron-deficient arylglyoxals - in these cases the "normal" reaction happens including the iminol rearrangement, but it is still valuable data.

Overall, this is a fairly complete study that extends our understanding of these multicomponent processes. The products are certainly somewhat limited to a specific kind of structure, but to a specialist audience, the results could be useful for those working on similar heterocyclic systems. I recommend publication after a few minor improvements, mostly related to presentation:

- The word "retarded" is perhaps not the most precise here - that rearrangement is not only slowed down but is completely uninvolved in the productive pathway, so maybe "avoided" would be a more suitable choice
- I am a bit worried about overinterpretation of the mechanistic data. In particular, there is quite a logical leap between the EPR experiment and the proposed role of cobalt, which in my opinion is not likely to be correct. If Co is acting here as a "deprotonation catalyst," the authors should try to be more specific about how it does so.
- To be clear, I think the results are publishable even if the authors do not figure out the exact (likely) role of the Co co-catalyst. It is OK to look more at whether the Co generates some base in situ by O₂ reduction or some other process, but if the conclusion is simply that the answer remains unknown, I think it is fine to be honest about that. The authors could certainly put a few proposals in the supplement and look into the process further in the future. The key mechanistic question in this study is really about slowing the 1,2-aryl migration, and I view the role of Co as somewhat secondary.
- The DFT calculations (note a few typos in spelling of DFT) do not provide any additional insight, in my opinion, and should be perhaps moved to the supplement to not interrupt the flow of the more important discussion.
- The methods and characterization data are OK. Can the authors comment on what represents the remainder of the mass balance when the yields are low?

Reviewer #2

(Remarks to the Author)

Wu, Xiang and co-workers report a novel approach to the synthesis of 2,3-diaryl-substituted indoles by intercepting a retarded α -iminol rearrangement with a Petasis reaction. The authors demonstrate that electron-withdrawing substituents on the aryl unit can slow down the rearrangement, allowing for intermolecular interception via boron-ate complex formation. This strategy enables regiospecific C(sp²)-C(sp²) bond formation using amines, arylglyoxals, and boronic acids, expanding the scope of Petasis-type reactions beyond traditional C(sp³)-C(sp²) linkages. The authors include experimental mechanistic studies and DFT calculations to support the proposed reaction pathway, which involves a copper-cobalt catalytic system.

The most interesting aspect of this work is the strategic interception of a retarded α -iminol rearrangement by a Petasis reaction, enabling the unprecedented regioselective synthesis of 2,3-diaryl-substituted indoles via a C(sp²)-C(sp²) bond-forming process. For that reason, I would like to recommend this manuscript for publication in Commun. Chem. if the authors can address the following issues:

- The proposed mechanism simplifies the role of Cu- and Co-catalysts too much. They should have more important roles in these reaction steps. The DFT calculations did not take the metal catalysts into account at all. The authors should try to come up with an improved detailed version of the mechanism.
- What will happen with aliphatic-substituted or non-substituted arylamine substrate(s)?
- There are a large number of typos and language issues not only in the text but also in graphics, the authors should perform a thorough proof-read before re-submission.

Reviewer #3

(Remarks to the Author)

Zhu et al. report the one-pot synthesis of 2,3-diarylindoles from an aryl amine, arylglyoxal, and arylboronic acid in the presence of a copper-cobalt catalyst. The key feature that permits this cascade reaction is the introduction of an electron-deficient aryl group on the glyoxal, such that a competing α -iminol reaction in the α -hydroxyiminium intermediate (I-A) is greatly decelerated, allowing for a Petasis reaction to occur instead. Considering that the competing borate complex formation is bimolecular, while the rearrangement is unimolecular, partitioning of I-A to the Petasis reaction requires a significantly electron-withdrawing para-nitro group, which destabilizes the phenonium ion intermediate presumed to form during the competing 1,2-migration. This study is an excellent example of how tandem reactions can be promoted by perturbing the electronics at a particular stage in the mechanism. I recommend publication after addressing a few minor points.

1. In several places in the manuscript (lines 22, 95, 191), the tandem reaction is described as triply convergent. The problem is that the phrasing requires a noun, not an adjective where convergent appears (e.g., the sentence concludes with "...this triply convergent."
2. Line 78: retard is the correct verb form.
3. Line 96: It is claimed that the previous experiments established optimum conditions, but these conditions are never explicitly stated. Presumably, the optimum conditions are the same as the "standard conditions", but this should be clarified.
4. Line 117: electronic, not electrical.
5. Figures 2 and 5 indicate that the position of R1 on the arylamine was varied, but it appears that in all cases, R1 is placed at the para position, and only a couple of examples (5c, 5d, 8o) include ortho or meta substituents. Clearly, a single meta substituent would lead to mixtures of regioisomers, but not in the case of a single ortho group. It would be helpful to clarify the substituent requirements for the arylamine and adjust the corresponding figures as appropriate.
6. Lines 95, 125: Check for consistent heading capitalization.
7. Line 160: approximately
8. Line 171 (two occurrences): DFT spelling (also, this should be written in full the first time it is used).
9. Line 184: delighted...slight
10. Line 187: insert a space between value and unit: 4 h
11. Line 195: rarely

Version 1:

Reviewer comments:

Reviewer #1

(Remarks to the Author)

The authors have thoughtfully responded to each of my suggestions. In my opinion, the manuscript is suitable for publication in its revised form.

Response to Manuscript ID: COMMSCHEM-25-0080A

(Other Version: COMMSCHEM-25-0080-T)

**Title: Intercepting a Retarded α -Iminol Rearrangement with a Petasis Reaction:
Synthesis of 2,3-Diaryl Substituted Indoles**

Dear Editor,

We are very grateful for the careful evaluation of our manuscript by the reviewers and the editorial office for constructive suggestions. We have thoroughly revised the manuscript following the reviewer's and the editor's comments. Manuscript files with and without highlighted changes have been uploaded as requested. Please find below the point-by-point response to these comments.

Note: The reviewer's comments are in **black**, and our responses are in **blue**.

Reviewer's comments:

Reviewer #1 (Remarks to the Author):

This manuscript by Wu and co-workers describes an interesting synthesis of 2,3-diaryl indoles using an aniline, a glyoxal, and a boronic acid. The authors have previously reported the synthesis of indolinones in a similar manner by condensation of anilines with glyoxals, followed by a typical 1,2-aryl shift to form the final carbonyl. Here, it is found that when the aryl group is very electron-deficient (p-nitro), that aryl shift is very slow, presumably due to disfavored phenonium formation, and the iminol intermediate is therefore long-lived enough to undergo Petasis-type addition to form the indole products that are the subject of the study. Some mechanistic discussion is provided, although the insight is somewhat limited by the complexity of the system, involving two first-row metal catalysts, and though the scope of the reaction is obviously limited, the authors do explore what happens with less electron-deficient arylglyoxals - in these cases the "normal" reaction happens including the iminol rearrangement, but it is still valuable data.

Overall, this is a fairly complete study that extends our understanding of these multicomponent processes. The products are certainly somewhat limited to a specific kind of structure, but to a specialist audience, the results could be useful for those working on similar heterocyclic systems. I recommend publication after a few minor improvements, mostly related to presentation:

-The word "retarded" is perhaps not the most precise here - that rearrangement is not only slowed down but is completely uninvolved in the productive pathway, so maybe "avoided" would be a more suitable choice

Response: Thanks to this reviewer's thoughtful suggestion, which we think is very reasonable, we have replaced "retarded" with "avoided" in the text.

-I am a bit worried about overinterpretation of the mechanistic data. In particular, there is quite a logical leap between the EPR experiment and the proposed role of cobalt, which in my opinion is not likely to be correct. If Co is acting here as a "deprotonation catalyst," the authors should

try to be more specific about how it does so.

Response: We believe that the reviewer's comments are very reasonable. We have changed the sentence: "An obvious signal captured when 5,5-dimethyl-1-pyrroline N-oxide (DMPO) was employed as a radical scavenger suggests that our reaction may involve the formation of radicals (Figure 3C) and those radical species may be related to the interaction between cobalt and O₂." into "An obvious signal captured when 5,5-dimethyl-1-pyrroline N-oxide (DMPO) was employed as a radical scavenger suggests that our reaction may involve the formation of radicals (Figure 3C)". We have also downplayed the discussion on the role of cobalt in the main text.

In addition, we added another DFT calculations in the Supporting Materials page S59. It illustrated that the process of forming boron-ate complex from a deprotonated hydroxyl group is 36.9 kcal/mol lower in energy than the process without deprotonation process in our reaction. This result suggests that it is favorable if the intermediate **I-A** loses a proton before it forms a boron-ate complex, rather than after. In conjunction with previous work [*Tetrahedron* **2010**, *66*, 2736–2745; *J. Org. Chem.* **2019**, *84*, 7950–7960; *J. Chem. Soc., Dalton Trans.*, **1997**, *24*, 4695–4700], the possibility of cobalt-assisted deprotonation is discussed in the Supporting Materials page S71.

-To be clear, I think the results are publishable even if the authors do not figure out the exact (likely) role of the Co co-catalyst. It is OK to look more at whether the Co generates some base in situ by O₂ reduction or some other process, but if the conclusion is simply that the answer remains unknown, I think it is fine to be honest about that. The authors could certainly put a few proposals in the supplement and look into the process further in the future. The key mechanistic question in this study is really about slowing the 1,2-aryl migration, and I view the role of Co as somewhat secondary.

Response: Thanks to the reviewer's suggestion, we changed our statement in the text into: *Given that cobalt acts primarily as a reaction accelerator rather than an essential additive (Table 1-6), we propose that the cobalt-oxygen synergy facilitates intermediate I-A deprotonation through in situ base generation via cobalt-mediated oxygen reduction.* We have also added sentences to the body of the text: "At this stage, we are still unable to accurately explain or prove this mechanism, especially the role of cobalt. We hope to further explore the mechanism of action of this bimetallic reagent in subsequent studies."

-The DFT calculations (note a few typos in spelling of DFT) do not provide any additional insight, in my opinion, and should be perhaps moved to the supplement to not interrupt the flow of the more important discussion.

Response: Based on the reviewer's suggestion, we have removed Figure 4 and related descriptions to the Supporting Materials.

-The methods and characterization data are OK. Can the authors comment on what represents the remainder of the mass balance when the yields are low?

Response: It is true that in most of our examples the products are of moderate yields. For the model reaction, we have tried our best to optimize the conditions, but the yields are still not

excellent. During our experiments we observed that the target product was the main product produced, accompanied by the production of many by-products in very low yields. We have tried to isolate and characterize the by-products and we believe that the following by-products might present in the system, each of which has a yield around 5% (R-Figure 1). We have also isolated some low yielding by-products whose structures could not be identified. We believe that the formation of these by-products is understandable. This transformation is a three-component reaction, there is a possibility that any two of the three substrates will react with each other under the (copper-catalyzed) aerobic oxidation conditions, thus leading to low yields of our target products (R-Figure 2).

R-Figure 1

R-Figure 2

(R1) *Journal of Molecular Catalysis A: Chemical* **2015**, *409*, 110-116

(R2) *Chem. Eur. J.* **2015**, *21*, 2954–2960

(R3) *Eur. J. Org. Chem.* **2023**, *26*, e202300620

Reviewer #2 (Remarks to the Author):

Wu, Xiang and co-workers report a novel approach to the synthesis of 2,3-diaryl-substituted indoles by intercepting a retarded α -iminol rearrangement with a Petasis reaction. The authors demonstrate that electron-withdrawing substituents on the aryl unit can slow down the rearrangement, allowing for intermolecular interception via boron-ate complex formation. This strategy enables regiospecific C(sp²)-C(sp²) bond formation using amines, arylglyoxals, and boronic acids, expanding the scope of Petasis-type reactions beyond traditional C(sp³)-C(sp²) linkages. The authors include experimental mechanistic studies and DFT calculations to support the proposed reaction pathway, which involves a copper-cobalt catalytic system.

The most interesting aspect of this work is the strategic interception of a retarded α -iminol rearrangement by a Petasis reaction, enabling the unprecedented regiospecific synthesis of 2,3-diaryl-substituted indoles via a C(sp²)-C(sp²) bond-forming process. For that reason, I would like to recommend this manuscript for publication in Commun. Chem. if the authors can address the following issues:

- The proposed mechanism simplifies the role of Cu- and Co-catalysts too much. They should have more important roles in these reaction steps. The DFT calculations did not take the metal catalysts into account at all. The authors should try to come up with an improved detailed version of the mechanism.

Response: We thank the reviewers for reviewing the manuscript and for their valuable comments, which we found very valuable. As we mentioned in the revision, we are not clear at this stage about the mechanistic processes and in particular the role of cobalt, and it is difficult to prove the role that cobalt plays. Therefore, we have emphasized the theme of the paper "slowing the 1,2-aryl migration" in the main text and downplayed the role of cobalt in the mechanism. Based on the available controlled experiments and the literature reported in the past, we believe that copper and cobalt do not undergo the classical process of metal insertion in our reaction system, but rather act as Lewis acids and bases to facilitate the reaction. We have tried to utilize DFT calculations to better account for the role of either copper or cobalt, but have not found a suitable explanation. In order to present the reader with a more continuous and clear description in the main text, we have placed more detailed mechanistic processes, especially possible mechanisms of action of cobalt, inside the Supporting Materials page S71.

- What will happen with aliphatic-substituted or non-substituted arylamine substrate(s)?

Response: Based on the reviewer's suggestion, we used N-benzyl-4-methoxyaniline and 4-methoxy-N-methylaniline as substrates for the reaction. and we succeeded in obtaining the target product (**5f**, 50%) and (**5g**, 62%), respectively. These experimental results illustrate that aliphatic-substituted arylamine can also be applied to this transformation. Furthermore, we used diphenylamine to carry out our reaction. Target product **5c** was obtained in only 20% yield.

- There are a large number of typos and language issues not only in the text but also in graphics, the authors should perform a thorough proof-read before re-submission.

Response: We have checked and corrected the text in detail throughout the text as well as in the images. We have highlighted all the changes in the manuscript. We hope that the reviewers

will be satisfied with our revised version.

Reviewer #3 (Remarks to the Author):

Zhu et al. report the one-pot synthesis of 2,3-diarylindoles from an aryl amine, arylglyoxal, and arylboronic acid in the presence of a copper-cobalt catalyst. The key feature that permits this cascade reaction is the introduction of an electron-deficient aryl group on the glyoxal, such that a competing α -iminol reaction in the α -hydroxyiminium intermediate (I-A) is greatly decelerated, allowing for a Petasis reaction to occur instead. Considering that the competing borate complex formation is bimolecular, while the rearrangement is unimolecular, partitioning of I-A to the Petasis reaction requires a significantly electron-withdrawing para-nitro group, which destabilizes the phenonium ion intermediate presumed to form during the competing 1,2-migration. This study is an excellent example of how tandem reactions can be promoted by perturbing the electronics at a particular stage in the mechanism. I recommend publication after addressing a few minor points.

1. In several places in the manuscript (lines 22, 95, 191), the tandem reaction is described as triply convergent. The problem is that the phrasing requires a noun, not an adjective where convergent appears (e.g., the sentence concludes with "...this triply convergent."

Response: Thanks to the reviewers' careful reading and revision, we changed "triply convergent" to "reaction" or "multicomponent reaction" in our revised manuscript.

2. Line 78: retard is the correct verb form.

Response: We have corrected this mistake accordingly.

3. Line 96: It is claimed that the previous experiments established optimum conditions, but these conditions are never explicitly stated. Presumably, the optimum conditions are the same as the "standard conditions", but this should be clarified.

Response: In the "Transformation developments" section, we have added a sentence explicitly stating that our "optimum condition" is specified as the "standard conditions".

4. Line 117: electronic, not electrical.

Response: We have corrected those mistakes accordingly.

5. Figures 2 and 5 indicate that the position of R1 on the arylamine was varied, but it appears that in all cases, R1 is placed at the para position, and only a couple of examples (5c, 5d, 8o) include ortho or meta substituents. Clearly, a single meta substituent would lead to mixtures of regioisomers, but not in the case of a single ortho group. It would be helpful to clarify the substituent requirements for the arylamine and adjust the corresponding figures as appropriate.

Response: Regarding this question from the reviewer, we have added two examples of 2,3-diaryl substituted indoles where the yield of the product is 20% when diphenylamine is used (5c, R1 = H) and 16% when di-o-tolylamine (5h, R1 = o-Me) is used. This latter result shows that the reaction is not good when the R1 on the arylamine is in the ortho position. Although the substrate is fully converted, we observed complex reaction outcome with low yields of the

desired product. We preliminarily believe that the substituents at this position will exhibit steric hindrance in the product.

6. Lines 95, 125: Check for consistent heading capitalization.

Response: We have corrected these mistakes accordingly.

7. Line 160: approximately

Response: We have corrected this mistake accordingly.

8. Line 171 (two occurrences): DFT spelling (also, this should be written in full the first time it is used).

Response: We have corrected these mistakes accordingly.

9. Line 184: delighted...slight

Response: We have corrected these mistakes accordingly.

10. Line 187: insert a space between value and unit: 4 h

Response: We have corrected this mistake accordingly.

11. Line 195: rarely

Response: We have corrected this mistake accordingly.

In addition, we have revised the full text and Supplementary Information according to the editor's revision requirements. Because of the additional products we added during the revision process as well as the extra DTF calculations, we have added a funding project number in the acknowledgements section. We have highlighted all the changes in the full text.

We thank again the referees for their thoughtful comments. We believe that the quality of the manuscript has been significantly improved after having addressed all these key questions raised by the reviewers.

Sincerely,

Prof. Jia-Chen Xiang and Prof. An-Xin Wu